# Treg Enhancing Therapies to Treat Autoimmune Diseases

**DOI:** 10.3390/ijms21197015

**Published:** 2020-09-23

**Authors:** Peter J. Eggenhuizen, Boaz H. Ng, Joshua D. Ooi

**Affiliations:** Centre for Inflammatory Diseases, Monash University Department of Medicine, Monash Medical Centre, Clayton, VIC 3168, Australia; peter.eggenhuizen@monash.edu (P.J.E.); boaz.ng@monash.edu (B.H.N.)

**Keywords:** Treg, Treg therapy, cell-based therapy, autoimmunity, autoimmune disease

## Abstract

Regulatory T cells (Tregs) are a small yet critical subset of CD4+ T cells, which have the role of maintaining immune homeostasis by, for example, regulating self-tolerance, tumor immunity, anti-microbial resistance, allergy and transplantation rejection. The suppressive mechanisms by which Tregs function are varied and pleiotropic. The ability of Tregs to maintain self-tolerance means they are critical for the control and prevention of autoimmune diseases. Irregularities in Treg function and number can result in loss of tolerance and autoimmune disease. Restoring immune homeostasis and tolerance through the promotion, activation or delivery of Tregs has emerged as a focus for therapies aimed at curing or controlling autoimmune diseases. Such therapies have focused on the Treg cell subset by using drugs to suppress T effector cells and promote Tregs. Other approaches have trialed inducing tolerance by administering the autoantigen via direct administration, by transient expression using a DNA vector, or by antigen-specific nanoparticles. More recently, cell-based therapies have been developed as an approach to directly or indirectly enhance Treg cell specificity, function and number. This can be achieved indirectly by transfer of tolerogenic dendritic cells, which have the potential to expand antigen-specific Treg cells. Treg cells can be directly administered to treat autoimmune disease by way of polyclonal Tregs or Tregs transduced with a receptor with high affinity for the target autoantigen, such as a high affinity T cell receptor (TCR) or a chimeric antigen receptor (CAR). This review will discuss the strategies being developed to redirect autoimmune responses to a state of immune tolerance, with the aim of the prevention or amelioration of autoimmune disease.

## 1. The Regulatory T Cell

Regulatory T cells (Tregs) are a subset of CD4+ T lymphocytes that possesses the capacity to suppress immune responses to ensure that the immune system’s response to foreign antigens and its response to self-antigens remains adequately balanced. This homeostatic balance ensures the immune system is inflammatory enough to respond sufficiently to foreign antigens, as well as neoantigens from cancer, and sufficiently counter-balanced or anti-inflammatory that inflammation does not get out of hand, leading to tissue damage or death. The immune-suppressive effects of Tregs regulate tumor immunity, antimicrobial resistance, allergy and transplantation [1,2,3,4]. Tregs also play a central role in maintaining self-tolerance. Treg involvement in this mechanism is pivotal for protection from autoimmune diseases [5,6].

### 1.1. Treg Phenotype

CD4+ T cells contain various pro-inflammatory and anti-inflammatory subsets, of which the Treg subset is the major anti-inflammatory subset. Tregs can be phenotypically distinguished from the other CD4+ T cell subsets by a high cell surface expression of the interleukin-2 receptor alpha chain (IL-2RA), CD25, as well as the transcription factor Forkhead box protein P3 (FOXP3), which has been shown to be critical for Treg development, function and stability [7,8]. Indeed, mutations in the *FOXP3* gene result in Immunodysregulation polyendocrinopathy enteropathy X-linked (IPEX) syndrome, characterized by Treg dysfunction and uncontrolled autoimmunity [9]. Treg phenotypic stability is best indicated by demethylation of the Treg-specific demethylated region (TSDR), a non-coding, evolutionarily conserved element within the gene locus of *FOXP3* [10]. In humans, but not in mice, Tregs are further distinguished from other CD4+ cells by a low expression of the IL-7 receptor CD127 [11]. These aforementioned markers have been critical in defining the Treg subpopulation, yet there are many other identifying markers which can assist in distinguishing Tregs beyond the scope of this review [12]. Such additional markers have a more variable expression, and for some, the mechanisms of their function are still being elucidated. The variation in Treg phenotype may depend upon the microenvironment or the target population to be controlled.

### 1.2. Treg Subsets

Tregs exist in two major subpopulations: natural Tregs (nTregs) and induced Tregs (iTregs). The iTregs can be further separated into subpopulations of IL-10-secreting CD4+ T regulatory 1 cells (TR1 cells), TGF-beta-secreting Treg cells (Th3) and CD8+ Treg cells [13]. nTregs arise from CD4+ single positive thymocytes, leave the thymus as FOXP3+ nTregs, and are enriched for T cell receptors (TCRs) that have a high affinity for self-peptides, thus playing an important role in autoimmunity [14]. Some CD4+ FOXP3- T conventional cells leave the thymus and, due to their phenotypic plasticity, are induced and converted into iTregs in the periphery after encountering an antigen, particularly in the presence of TGF-beta and IL-2 [15]. The phenotypic plasticity of iTregs permits an immune-tolerant state in extreme inflammatory conditions and further assists nTregs in restoring immune tolerance when needed [16,17].

### 1.3. Treg Function

Tregs effect immune tolerance by direct and bystander suppression. In direct suppression, Tregs suppress the target cell in an antigen-specific manner [18,19]. In bystander suppression, Tregs specific for one antigen have the ability to suppress immune responses against other antigens due to their close proximity to the antigen-specific response. This anti-inflammatory response restores immune tolerance and maintains immune homeostasis [20]. Treg suppressive function occurs via cell-to-cell crosstalk mechanisms, which are contact-dependent, as well as the secretion of inhibitory cytokines such as IL-10, IL-35 and TGF-beta, which have anti-inflammatory signalling properties [14]. Tregs also suppress immunity by acting as an IL-2 sink; as they possess a high surface expression of IL-2 receptor (CD25), they can soak up extracellular IL-2, thereby dampening pro-inflammatory cytokine signalling. IL-2 is also critical for Treg functional activity and survival [21,22]. Antigen-specific direct suppression occurs when peptide-loaded MHC class II (MHCII) on antigen-presenting cells (APCs) engages with the TCR complex on the Treg. If the TCR is able to recognise this peptide-MHC, the TCR complex undergoes a signal transduction cascade, resulting in Treg activation and an increase in Treg suppressive function.

### 1.4. T Cell Receptor

The T cell receptor, present on all T cells, confers antigen-specificity to the T cell through interaction with its cognate peptide-MHC on APC, permitting it to activate in an antigen-specific manner. On Tregs, the TCR exists as an alpha-beta heterodimer that is restricted to antigens presented by MHCII [23]. There is a great diversity in the specificity of TCRs due to somatic V(D)J recombination in the thymus, which has the capacity to generate in humans up to 10^61^ different TCR sequences [24]. High affinity TCRs on Treg cells are known to elicit a strong response to self-peptides as a result of thymic T cell selection selecting for Tregs with high-affinity, self-peptide recognising TCRs. Conversely, CD4+ T conventional cells (Tconvs) possess TCRs that have a lower affinity for their cognate peptide-MHC, are more flexible in their cross-reactivity, and possess a higher specificity to foreign peptides, which aligns with their T effector functions [25,26,27]. After recessive immune tolerance deleting most self-reactive TCRs in the thymus, a subset of those that remain obtain FOXP3 expression and nTreg phenotype, which is promoted through the autoimmune regulator transcription factor AIRE [28,29,30,31]. Therefore, the TCR repertoire from thymically-derived nTregs differs from that of Tconvs. TCR signalling on Tregs is required for their activation and ability to suppress [18,19,32]. When TCR engages with peptide–MHCII in the periphery, it is able to activate the Treg by signal transduction through the TCR–CD3 complex. Compared to Tconvs, Tregs have impaired TCR signal transduction, referred to as TCR hyposignalling [33]. The signalling components of the TCR signal transduction cascade between Tconvs and Tregs are significantly different in their phosphorylation status, but not in their abundance or composition, except for Themis which has been detected at lower levels in Tregs. [34]. Compared to Tconvs, Tregs have altered mechanisms of co-receptor binding involving the TCR–CD3 complex. Lymphocyte Activation Gene 3 (LAG3) on Tregs is a homologue of CD4, and binds to MHCII at a higher affinity than CD4. This results in the further enhancement of Treg signalling and contributes to their suppressor activity [35]. Treg costimulation is different to that of Tconvs by constitutive surface expression of CTLA-4, a key inhibitory molecule dependent on FOXP3 that competes with the major costimulatory molecule, CD28. CTLA-4 is able to outcompete CD28, binding to B7 family members B7-1 (CD80) and B7-2 (CD86) on APCs [36]. CTLA-4 possesses additional suppressor functions by disrupting the location of CD28 at the immune synapse, resulting in a shortened dwell time between naïve T cells and APCs. CTLA-4 also inhibits TCR signalling through recruitment of phosphatases SHP-2 and PP2A [37,38]. Once the Treg is activated, it can broadly suppress by means of direct antigen-specific immune tolerance or bystander suppression, which enables the blocking of the responses of nearby cells. Not only does the TCR control the antigen-specificity of the response but, additionally, the affinity of a TCR for a given peptide–MHC complex determines the potency of the Treg response [39]. Therapies exploiting the TCR to elicit an enhanced, autoantigen-specific Treg activity and function are currently being explored as a cell-based therapy in autoimmune disease, and will be discussed in detail later.

## 2. Autoimmune Disease

Autoimmune diseases, such as multiple sclerosis (MS), type 1 diabetes (T1D) and systemic lupus erythematosus (SLE), are caused by chronic immune responses against the host’s own cells, tissue and organs, which subsequently culminates in tissue destruction, dysfunction and pathology. These autoimmune disorders develop from a set of poorly-defined interactions between environmental triggers and polymorphic genomic elements that trigger a loss of self-tolerance. The complexity and heterogeneity of the autoimmune response has proved challenging with regard to the development of targeted treatments, as they are required to sufficiently purge the immune system of autoreactivity while maintaining the normal functional side of the immune system [40]. Indeed, many current treatments for autoimmune diseases fail in this regard, as they are not specific to the autoimmunity, but dampen immune responses systemically. The current standard of care for most autoimmune diseases is broadly immunosuppressive drugs, such as corticosteroids. These broad immunosuppressants carry significant side effects, including risking severe infections and potentiating cancer. Biologicals, usually antibodies targeting one component of the immune system, are starting to be used to treat certain groups of autoimmune disease patients. This is a promising step towards safer therapies, however the efficacy of biologicals is still being assessed [41,42].

### Enhancing Treg Immunosuppression to Treat Autoimmune Disease

There are a number of therapies for autoimmune diseases being explored, which are focused on enhancing the anti-inflammatory or tolerogenic component of the immune system (Figure 1). The aim of these therapies is to control autoimmunity such that immune tolerance is restored. Recent drug-based targets for ameliorating autoimmunity have focused on molecules to promote the in vivo induction and expansion of Tregs. Drugs such as the mTOR-inhibitor rapamycin, or biologicals like the administration of IL-10, low-dose IL-2, TNF receptor 2 (TNFR2) agonists or FMS-like tyrosine kinase 3 ligand (Flt3L), have been explored [21,43,44,45,46,47]. The administration of autoantigens has the potential to restore tolerance or vaccinate against autoimmunity. Numerous modes of autoantigen administration have been trialled in mice and humans, such as by DNA vaccination, leading to transient expression of the autoantigen, and oral, intra-nasal, sub-cutaneous or intravenous administration of autoantigens [48,49]. More sophisticated approaches of autoantigen delivery to treat autoimmune disease have involved coupling the autoantigen to a variety of vectors to deliver the autoantigen. One such approach involves the use of peptide–MHC-coupled nanoparticles to deliver the autoantigen, promoting tolerogenic dendritic cells (DCs) and the expansion of Tregs [50,51]. Cell-based therapeutics are emerging as candidates for the treatment of autoimmune disease. Approaches have utilized expanded TR1 cells expressing IL-10 [52], Tregs expressing a natural repertoire of polyclonal TCRs [53] or Tregs that have been ex vivo-engineered to express a receptor, such as a TCR, that is specific for the auto-antigen exacerbating the autoimmune disease or a chimeric antigen receptor (CAR) [54,55]. DCs also offer a cell-based therapeutic pathway to restore tolerance and prevent autoimmunity. Tregs can generate a tolerogenic phenotype in DCs, which can contribute to the restoration of immune tolerance. Since tolerogenic DCs are able to suppress autoimmunity, administering ex vivo-generated tolerogenic DCs is being explored [56]. Other approaches have used mature DCs to expand antigen-specific Tregs [57]. A wide variety of therapeutic approaches enhancing the anti-inflammatory aspect of the immune system are being explored to treat autoimmune disease. This offers an opportunity to improve on current treatment regimens, with subsequent improvements in patient outcomes.

## 3. Non-Cell-Based Therapies

### 3.1. Low-Dose IL-2

The cytokine IL-2 plays a crucial role as a pro-inflammatory signalling molecule in response to microbial infection. It has an added role in controlling autoimmunity through Tregs. The discovery that mice lacking IL-2 exhibit uncontrolled T cell activation and rampant autoimmunity explained just how crucial IL-2 is for the development, homeostasis and suppressive function of Tregs [58,59]. Tregs have a high, constitutive surface expression of CD25, the IL-2 receptor alpha chain. However, other immune cells such as natural killer (NK) cells, other T cells and some innate lymphoid cells also express CD25, albeit at lower levels compared to Tregs. IL-2 has the potential to expand Tregs in vivo in autoimmune disease patients [21]. Therefore, the basis for using a low dose of IL-2 as a therapeutic for autoimmune disease was hypothesized. By using a low dose, the IL-2 preferentially targets the high CD25-expressing Tregs [60]. Such approaches have been used in the context of infection and transplant with graft versus host disease (GVHD) [61,62]. In the context of autoimmunity, low-dose IL-2 has been shown to be safe in T1D and SLE [63,64,65,66]. Further developments utilizing the Treg-promoting effects of IL-2 involve using IL-2–anti-IL-2 complexes to expand Tregs in vivo, as well as engineering antigen-specific Tregs to express an altered IL-2 receptor that would only bind to an altered IL-2, which, when administered, would selectively expand only the antigen-specific Treg subset [67,68]. An advantage of IL-2 therapy is that recombinant human IL-2 is already available in the clinic as a therapeutic drug called Aldesleukin or Proleukin, for the treatment of malignant melanoma and renal cell carcinoma [69].

### 3.2. Rapamycin

The mTOR inhibitor rapamycin is a potent macrolide immunosuppressant drug, which blocks signalling in response to cytokines or growth factors, blocks T cell cycle progression and promotes TCR-induced T cell anergy and deletion [70]. In DCs, it inhibits their ability to mature and stimulate Tconvs [46,71]. In contrast to Tconvs, rapamycin selectively expands Tregs while maintaining their suppressive phenotype and function. Altogether, the effect of rapamycin results in an induction and enrichment of Tregs, which helps to promote immune tolerance [43,44,45]. The tolerance-restoring effects of rapamycin have been shown in T1D in mice. This effect was enhanced with the administration of IL-10 [45]. Flt3L, in a synergistic combination with rapamycin, has also been shown to further boost the induction of Tregs and their ability to promote tolerance through the selective expansion of plasmacytoid DCs. This works by rapamycin blocking the expansion of the conventional DCs but not the plasmacytoid DCs. An increase in mTOR activity in plasmacytoid DCs makes their Flt3L signalling pathway more resistant to rapamycin. This synergistic effect of plasmacytoid DCs, rapamycin and Flt3L in inducing Tregs seems to only be applicable as a combination therapy, since Flt3L alone fails to induce Tregs [46]. Therefore, rapamycin is a suitable candidate for the in vivo induction of tolerance, and it would be worthwhile to assess the combined effect in adoptively transferred antigen-specific Treg therapy.

### 3.3. TNF Receptor 2

TNFR2 is a receptor for the pro-inflammatory cytokine TNF-alpha, which can induce anti-inflammatory and tissue regenerative effects. This is in contrast to TNF receptor 1, which also binds to TNF-alpha but with a pro-inflammatory effect. It is known that Tregs have a higher expression of TNFR2 compared to other T cell subsets, and its expression is linked to Treg-suppressive capability in mice and humans [47,72,73]. TNFR2 agonism in the context of autoimmunity has been shown in T1D. In an animal model for T1D and ex vivo human studies from T1D patients, TNFR2 agonism was able to selectively kill autoreactive CD8+ cells [74,75]. TNFR2 agonism has not reached clinical trials for autoimmune disease. Given the role of TNFR2 in controlling inflammation, targeting this receptor may be of benefit in the context of autoimmunity.

### 3.4. Peptide Administration

Antigen-specific Tregs can be activated and expanded through administration of autoantigens, and can be regarded as a type of vaccine against autoimmunity. In mouse models, tolerance to these antigens has been demonstrated across various autoimmune diseases, such as experimental autoimmune encephalomyelitis (EAE), rheumatoid arthritis (RA), T1D and ANCA-associated vasculitis [48,76]. There have been multiple approaches to the delivery of the autoantigen of interest, such as oral, intranasal, systemic, intramuscular, subcutaneous and intravenous. Methods using various techniques, such as autoantigenic peptides, whole antigen, autoantigen-loaded anti-DEC205, peptide–MHC multimers, autoantigen-loaded erythrocytes and autoantigen loaded ECDI-fixed mononuclear cells, have been explored [40]. Other approaches include loading autologous tolerogenic DCs with autoantigens, which can induce the differentiation and expansion of Tregs [48]. Another approach involves engineered DNA plasmids encoding the autoantigen being intramuscularly injected, taken up by tolerizing DCs and promoting autoreactive T effector cell suppression or their differentiation into Tregs [49]. Others have administered such DNA vectors via the liver, utilizing the liver’s tolerogenic effects [77,78]. Mouse studies into this have promoted Treg expansion and disease reversal in a mouse model of diabetes [79]. Additionally, nanoparticles coated with autoantigenic peptides bound to MHC class II molecules have been shown to trigger the generation and proliferation of antigen-specific TR1 and Treg cells in mouse models, leading to the restoration of immune tolerance [50,51]. Dozens of clinical trials involving DNA vaccines, peptides, proteins and epitopes loaded onto relevant carriers have been initiated, and most were shown to be safe and well tolerated [40]. However, the findings in mice and in clinical trials have not yet been translated into therapeutics for humans, owing to the complexity behind autoimmunity and the paucity in understanding its mechanisms. Overall, peptide administration alone or in combination with other Treg-inducing approaches may protect from or reverse autoimmunity. Further understanding of the mechanisms underlying autoimmunity and the development of autoantigen administration are necessary in order to evaluate the efficacy of autoantigen administration in humans.

## 4. Cell-Based Therapies

### 4.1. Microbiome Therapy

A different strategy for promoting Tregs and the pathways involved in immune tolerance and homeostasis is performed through the microbiome. During chronic inflammation, such as that experienced during autoimmune disease, the composition of the microbiome is drastically changed, which can add to the imbalance of immune dysregulation [80]. Fecal microbiome transplantation involving specific bacteria may be a supplement to other therapies to combat autoimmune disease. Short chain fatty acids, such as butyrate, are produced by certain bacterial species in the gut. Butyrate has been shown to have an anti-inflammatory effect, capable of upregulating anti-inflammatory genes in DCs, enhancing histone acetylation of the FOXP3 locus and improving the stability of the FOXP3 transcription factor [81]. Other molecules produced by bacteria, such as cell surface polysaccharies beta-glucan/galactan and polysaccharide A from *Bifidobacterium bifidum*, are able to induce Tregs [82]. For safety reasons, fecal transplantation therapy is not currently approved for clinical use. When available, it could be considered as a putative Treg-promoting therapy to use in conjunction with Treg cell-based therapies or other Treg-promoting therapies to enhance their effect.

### 4.2. Treg Therapy

Autoantigen-specific Treg dysfunction, in terms of decreased cell number and decreased suppressive phenotype, has been reported to exacerbate autoimmune diseases [25,83]. Therefore, in order to restore dominant immune tolerance, the administration of Treg cells is hypothesized to restore the Treg imbalance by directly increasing Treg cell numbers, leading to heightened immune suppression (Figure 2). The adoptive transfer of polyclonal Tregs has been shown to be safe and efficacious in T1D [53]. Treg therapy can be further enhanced by engineering the cells to be autoantigen-specific, which will increase their potency and suppressive effect [84]. For the therapy to be successful, one must ensure Tregs remain specific and stable, and can survive long-term and maintain their suppressive capacity. The first animal in vivo evidence was shown 30 years ago by Hall et al., who showed that adoptively transferred CD25+ T suppressor cells can maintain tolerance in rat transplantation models [85]. Over 20 years ago, ground-breaking studies by Groux et al. showed that in vitro expanded IL-10-producing TR1 cells improved colitis in mouse models [86]. This showed that an antigen-specific CD4+ subset was able to inhibit antigen-specific T cell responses and prevent colitis in vivo. The idea of adoptively transferring antigen-specific T cells to treat autoimmune disease was taken further by Desreumaux et al. in 2012 [52]. They published the first in-human results of adoptively transferred, OVA-specific TR1 cells, actively improving Crohn’s disease. The treatment was safe and efficacious. Since then the field has broadened, exploring a variety of approaches to using Tregs as a living therapy in the treatment of autoimmune disease, although there still remains a number of challenges to overcome before the therapy can be adopted into clinical practice.

### 4.3. Polyclonal Treg Therapy

Polyclonal Treg therapy uses autologous ex vivo-expanded Tregs to restore tolerance in autoimmune diseases. A number of clinical trials using polyclonal Treg therapy for the treatment of autoimmune disease have been completed or are ongoing (Table 1). The first such clinical trial (ISRCTN06128462) in children with T1D, which showed adoptive transfer of polyclonal, autologous Tregs, prolonged the survival of pancreatic islets. Some patients showed signs of clinical remission and remained insulin-independent after one year without any adverse effects [87]. Another clinical trial (NCT NCT01210664) using polyclonal autologous ex vivo-expanded Tregs, undertaken in adult T1D patients, showed the treatment is safe and tolerable [53]. Positive results from these clinical trials have allayed concerns that polyclonal Treg therapy would promote generalized immune suppression, leading to an increased risk of infection and cancer, which has not been found. Further clinical trials using polyclonal Tregs as a cell-based therapeutic are currently underway in T1D and other autoimmune diseases to explore the best treatment regimen, in terms of cell numbers transferred, number of doses required and time between doses. Other clinical trials using polyclonal Tregs in the treatment of autoimmune disease include for the treatment of active cutaneous pemphigus (NCT03239470), autoimmune hepatitis (NCT02704338) and Crohn’s disease (NCT03185000). One mixed result from a phase I clinical trial (NCT02428309) using polyclonal Tregs in SLE found that the administered Tregs were able to traffic to the site of autoimmunity, in this case the skin, yet there was an increase in pro-inflammatory IL-17 from CD4+ and CD8+ cells [88]. It is noteworthy that only one patient was tested in this trial. Perhaps the use of other Treg-promoting therapies in combination with Treg therapy would boost the suppressive function and number of Tregs with improved patient outcomes, such as rapamycin, IL-10, IL-2, TNFR2, nanoparticles, etc. One such phase I clinical trial (NCT02772679) is looking at co-administration of IL-2 with polyclonal Tregs. As mentioned earlier, low-dose IL-2 therapy alone has the effect of expanding in vivo Tregs. Co-administration of polyclonal Tregs and low-dose IL-2 is expected to boost Treg number and function after administration. Indeed, the combination therapy of Tregs with other therapies may also have a boosting effect in the treatment of autoimmune disease, and such avenues remain to be explored. The immunosuppressive regimen that patients are on also needs to be taken into consideration when using Treg therapy. Immunosuppressant drugs commonly used to treat autoimmune disease, such as methylprednisolone, mycophenolate and tacrolimus, all have been shown to reduce Treg proliferation and viability in a dose-dependent manner [89]. Therefore, careful consideration in tailoring an appropriate treatment regimen when undergoing Treg therapy is essential.

### 4.4. Engineered Antigen-Specific Treg (TCR-Treg) Therapy

Treg therapy can be enhanced by the introduction of an autoantigen-specific TCR (TCR-Treg). This has the ability to redirect their response towards a desired autoantigen specificity. Tregs can be ex vivo-transduced, by way of retroviral or lentiviral transduction, to express a high-affinity, autoantigen-specific TCR, and can be subsequently expanded for use as a cell-based therapy to treat a specific autoimmune disease. A seminal discovery in a mouse model for T1D showed that NOD mice engineered to express a diabetogenic TCR required only a low number of autoantigen-specific Tregs to sufficiently prevent, and in some cases even reverse, T1D [90]. Noteworthy from this study was that polyclonal Tregs were not able to prevent or reverse disease in the model. These TCR–Tregs were able to be expanded ex vivo and be used as a cell-based therapy to cure T1D in mice. Another group showed that as few as 2000 antigen-specific Tregs were all that was required to prevent T1D in mice [91]. A lower dosage makes the antigen-specific Treg approach for cell based therapy advantageous compared to polyclonal Tregs, and the optimal dosage and timing still needs to be determined in humans. Although very few antigen-specific Tregs may be required to ameliorate autoimmune disease compared to polyclonal Tregs, the identification of an appropriate, high-affinity, autoantigen-specific TCR for transduction onto a Treg remains a challenge for some autoimmune diseases with poorly defined dominant epitopes. The great diversity in TCRs, coupled with there being very few antigen-specific Tregs naturally circulating in the peripheral blood with most of them residing in tissues, makes them hard to isolate and identify, and has proved to be a challenge in identifying appropriate TCRs for use in therapy. Single-cell sequencing is required for TCR identification, since each T cell clone expresses a different TCR sequence to the other, and the successful sequencing of both the alpha and beta chain TCR is required to successfully identify one TCR [92]. The technical challenges of single-cell sequencing are nowadays being overcome by advances in single-cell, whole-transcriptome, next-generation sequencing. Such technologies have the capacity to sequence and identify the TCRs alongside the whole transcriptome and TCR–peptide–MHC dextramer binding data of tens of thousands of single cells simultaneously [93,94]. Numerous TCRs have been used to generate human TCR-Tregs to target various autoimmune diseases, such as T1D, MS and acquired factor VIII deficiency [95,96,97].

Improvements in transduced TCR design have been explored. TCR mispairing may occur between the alpha-beta TCR chains of the endogenous TCR and the alpha-beta TCR chains of the transduced TCR. TCR mispairing has been shown to be reduced by a number of methods, such as by endogenous TCR gene silencing, CRISPR-mediated endogenous TCR gene deletion, or modifications on the transduced TCR, such as cysteination of the alpha-beta constant chains forming a disulphide bond to promote correct alpha-beta chain TCR pairing [98,99,100,101,102].

The advantageous effect of Tregs transduced with an auto-antigen specific, high-affinity TCR is their increased potency in suppressive function compared to Tregs transduced with a non-specific TCR or polyclonal Tregs. These more potent Tregs are better equipped to restore dominant immune tolerance, leading to complete remission of the targeted autoimmune disease [95].

### 4.5. Chimeric Treg Therapy

An MHCII-restricted TCR is not all that can be transduced into a Treg to enhance its function or specificity. Theoretically, any protein of interest can be transduced into a Treg. A great variety of creative chimeric Tregs have already been designed and shown to be effective in models of autoimmune disease. In a murine model of MS, a myelin peptide–MHCII complex was fused to the CD3ζ cytoplasmic tail in Tregs. This promoted their activation when in contact with cognate effector T cells [103,104]. This enhancement was further developed by transducing a FOXP3 transgene along with the peptide–MHCII complex into T conventional cells [105]. MHC class I-restricted TCR has also been transduced onto a Treg cell and shown to maintain the antigen-specific suppressive capacity of the transduced TCR [106]. This showed that TCRs from cells other than Tregs, such as CD8+ T cells or CD4+ Tconvs, could be transduced onto Tregs. Further development of this technology is required before it can be translated to human therapy.

### 4.6. CAR vs. TCR Treg Therapy

Another example of a chimeric Treg is the CAR Treg, which possesses a CAR molecule comprising an extracellular, antigen-recognition domain based on antibody variable domains, and a transmembrane region followed by an intracellular, ITAM-rich signalling domain based on T cell signalling machinery. CAR Tregs have the ability to bind to tissue-specific autoantigens and traffic to the site of autoimmunity, specifically focusing their suppressor functions at the diseased site [107]. A caveat of this is the autoantigen being solely expressed at the site of autoimmunity. If the autoantigen is expressed elsewhere in the body in healthy tissue, the antigen-specific response may not be as effective, as it may result in a systemic hyper-activation of the CAR Tregs, leading to side effects such as general immune suppression [108]. CAR T cells have been used in the context of autoimmunity in mouse models of pemphigus to specifically target and kill autoantigen-specific B cells [109]. Focusing Treg therapy on one specific epitope for an autoantigen may not adequately address autoimmune diseases which have a large autoantigenic repertoire of T or B cells, or where epitope spreading has occurred. CAR Tregs have a greater affinity for their cognate antigen than TCR Tregs, which may be of benefit. However, CARs require the target cell to have at least 100 target autoantigens for the CAR to successfully recognise it and stimulate the Treg [110]. This is contrasted with a TCR Treg, which requires only one peptide–MHC interaction with TCR in order for the Treg to become activated [111]. This difference needs to be considered alongside varying levels of surface expression, as TCRs are typically present at approximately 50,000 per cell, whereas CARs have a variable expression but a greater magnitude of >50,000 per cell [54]. CAR Tregs possess a greater signalling ability than TCR Tregs, due to a greater number of ITAMS and tyrosines as substrates in the intracellular signalling domain leading to a stronger level of activation [54]. TCRs are heterodimers, whereas CAR is a monomeric protein, making expression less complex. TCRs require co-receptor involvement for co-stimulation, such as CD4 and CD28, whereas CAR does not require such co-receptors for stimulation. TCRs are always restricted to a particular MHC allele, and therefore multiple TCRs would need to be identified to cover the variety of MHC alleles in the diseased population. Given the advantages and disadvantages of TCR Tregs and CAR Tregs, further research needs to be undertaken to provide insight into which technology is more appropriate for a particular application.

### 4.7. DC Therapy

Dendritic cells, the body’s professional APCs, play a central role in the initiation and control of immune responses (Figure 3). They regulate the balance between immunity and tolerance. Tolerogenic DCs are the population of DCs that are essential in maintaining peripheral and central tolerance through the induction of clonal T cell anergy and deletion, the inhibition of T cell memory and Tconvs, and the activation and proliferation of Tregs [112]. Mature DCs arise in inflammatory environments, secrete pro-inflammatory cytokines, such as IL-1 beta, IL-12, IL-6 and TNF, and have a high co-stimulatory surface expression of CD80 and CD86, as well as high MHCII expression, all of which stimulate Tconvs as well as being able to induce Tregs [113]. Tolerogenic DCs exhibit an immature or semi-mature phenotype, which is characterized by lower expressions of CD40, CD80, CD86 and MHCII, and the secretion of anti-inflammatory cytokines, such as high levels of IL-10 and TGF-beta and low levels of IL-12 [112]. Tolerogenic DCs also secrete retinoic acid and indoleamine 2,3-dioxygenase (IDO), both of which enhance Treg survival and proliferation [114,115]. The phenotypic signature of tolerogenic DCs promotes the induction and proliferation of Tregs, which can be harnessed as a cell-based therapy.

Cell-based therapies utilizing the Treg-promoting effects of dendritic cells offer a way to expand antigen-specific Treg cells in the context of autoimmunity (Figure 3). Studies in mice have shown that mature DCs are able to expand antigen-specific Tregs, and these Tregs exert a greater antigen-specific suppressive capability [57]. Tolerogenic DCs are a subset of DCs that possess immunosuppressive properties that are able to prime the immune system into a tolerogenic state. Ex vivo-generated tolerogenic DCs have been shown in a phase I clinical trial of T1D to be safe and well tolerated, as well as showing some clinical signs of B cell suppression [56]. There are also a number of clinical trials underway using tolerogenic DCs in the treatment of autoimmune diseases, such as T1D, rheumatoid arthritis, MS and Crohn’s disease [109,116]. For tolerogenic DCs to be a successful cell-based therapy, more information on in vivo stability and phenotype needs to be compiled. Additionally, optimization for the best mode of delivery and the best dose of tolerogenic DCs remains to be determined. Adopting a standardized ex vivo tolerogenic DC generation method would also be beneficial.

## 5. Conclusions

There is currently a spectrum of new therapeutic approaches aimed at promoting the effects of Tregs to combat debilitating autoimmune diseases. Many of these new biologicals and cell-based therapies are already in clinical trials, but it remains to be seen which novel therapeutic approaches will eventually be adopted in the clinic. Perhaps a combination of aforementioned therapies will promote an enhanced targeted effect and improve outcomes for patients. Cell-based therapies offer an exciting new realm of treatments for autoimmune disease. Phase I and II clinical trials of polyclonal Tregs in T1D and other autoimmune diseases show that this therapy is safe and efficacious, and investigations into Treg therapy for other autoimmune diseases should be undertaken. Coupling the potent anti-inflammatory effects of Tregs with antigen specificity by transduction of a specific receptor offers much hope for future therapies to be even more specific and efficacious. The development and production of a successful Treg therapy continues to represent an exciting and challenging endeavor, which ultimately may improve autoimmune disease patient outcomes.

## Figures and Tables

**Figure 1 ijms-21-07015-f001:**
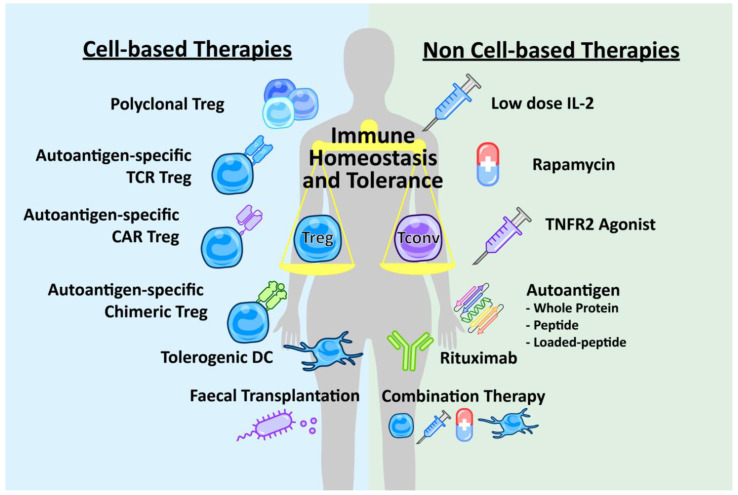
New therapeutic approaches for autoimmune disease. A variety of cell-based therapies and non-cell-based therapies are currently being explored in the treatment and prevention of autoimmune diseases. There are cell-based therapies, such as ex vivo-expanded polyclonal Tregs or Tregs transduced with an autoantigen-specific T cell receptor (TCR), chimeric antigen receptor (CAR), or other chimeric receptors such as peptide-MHC. Another cell-based approach utilises the immune-regulating effects of tolerogenic dendritic cells (DC). Faecal transplantation of specific Treg-promoting bacteria may also be beneficial. Non-cell-based therapies include biologicals such as low-dose IL-2 therapy, TNF receptor 2 (TNFR2) agonist therapy and the anti-CD20 antibody rituximab. Drugs, such as rapamycin, can also promote Treg proliferation. Administration of the autoantigen by various methods can vaccinate against autoimmunity. Finally, a combination of such therapies may be more efficacious than one therapy alone.

**Figure 2 ijms-21-07015-f002:**
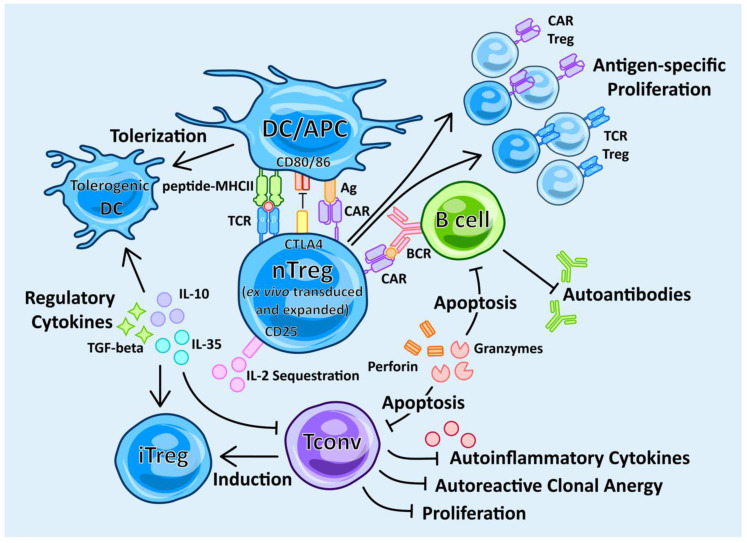
Mechanisms of action in Treg therapy. Tregs as a cell-based therapy work in many ways to restore immune tolerance. They secrete anti-inflammatory regulatory cytokines, such as IL-10, IL-35 and TGF-beta, which promote the induction of Tconvs into iTregs and the tolerization of dendritic cells (DC) to tolerogenic DCs. Tregs act as a sink for IL-2 through the cell surface IL-2 receptor, CD25. Tregs can directly kill inflammatory cells, such as Tconvs and B cells, by the secretion of perforin and granzymes. Tregs can suppress autoreactive B cells and autoantibody production, as well as suppressing autoreactive Tconvs. Treg therapy can be non-specific by the use of ex vivo-expanded polyclonal Tregs expressing a natural repertoire of T cell receptors (TCRs). By transducing either an antigen-specific TCR or chimeric antigen receptor (CAR) onto the Treg, the Treg can be redirected to specifically activate and target the auto-antigenic immune response. Such antigen-specific interactions increase the potency of the Treg response mechanisms in response to the autoantigen of interest, as well as promoting a clonally expanded set of transduced Tregs to further curtail autoimmunity.

**Figure 3 ijms-21-07015-f003:**
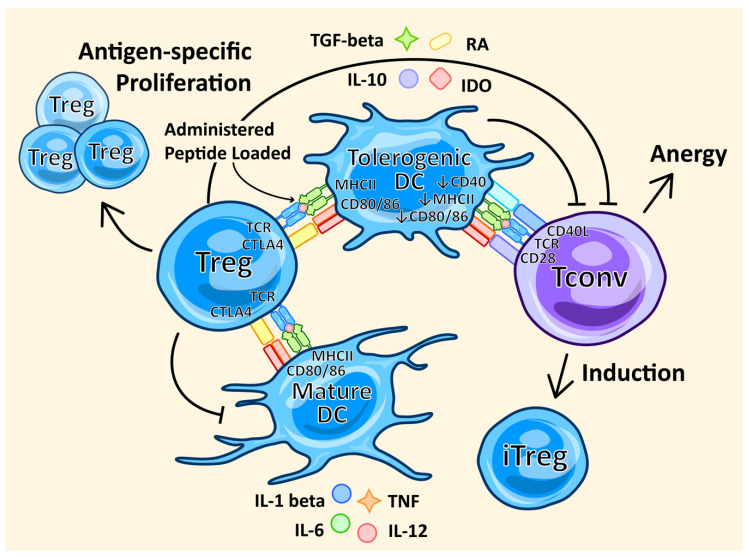
Dendritic cells (DCs) promote Tregs. Mature DCs have the ability to promote Treg proliferation, which can be directed to be antigen-specific. In turn, Tregs have the ability to tolerize DCs, affording them an immunosuppressive phenotype. Tolerogenic DCs’ immunosuppressive mechanisms include decreased surface expression of MHCII and co-stimulatory molecules CD40, CD80 and CD86, and the secretion of indoleamine 2,3-dioxygenase (IDO), retinoic acid (RA) and anti-inflammatory cytokines IL-10 and TGF-beta, all of which result in tolerogenic DC induction, inhibition of Tconv effector function, Tconv anergy, and iTreg induction.

**Table 1 ijms-21-07015-t001:** Clinical trials using Tregs to treat autoimmune diseases.

Study ID	Phase	Disease	Therapy	Status/Outcome
ISRCTN06128462	I	T1D	Autologous polyclonal expanded nTregs	Completed. Safe and well tolerated. Most patients responded to the therapy. Patients required lower amounts of exogenous insulin. Autoantibody status unchanged.
NCT02691247	II	T1D	Autologous polyclonal expanded nTregs	Completed. Well tolerated. No improvement in C peptide levels after one year.
NCT02772679	I	T1D	Autologous polyclonal expanded nTregs & IL-2	Active, not recruiting
NCT03011021	I/II	T1D	Umbilical cord blood polyclonal expanded nTregs	Recruiting
NCT01210664	I	T1D	Autologous polyclonal expanded nTregs	Completed. Safe and well tolerated. C peptide levels remained after 2 years. No change in autoantibodies
NCT03444064	I	T1D	Autologous polyclonal expanded nTregs	Recruiting
NCT02932826	I/II	T1D	Umbilical cord blood polyclonal expanded nTregs	Recruiting
NCT02704338	I/II	Autoimmune hepatitis	Autologous polyclonal expanded nTregs	Unknown
NCT02327221	II	Crohn’s Disease	In vitro differentiated and expanded autologous TR1 cells specific for OVA	Adverse events in a few patients. Remission was associated with lower dose of TR1 cells
NCT03185000	I/II	Crohn’s Disease	Autologous polyclonal expanded naive nTregs	Not yet recruiting
NCT02428309	I	Cutaneous lupus	Autologous polyclonal expanded nTregs	Terminated due to participant recruitment feasibility
NCT03239470	I	Pemphigus	Autologous polyclonal expanded nTregs	Active, not recruiting

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
