# Peer review of "Treg Enhancing Therapies to Treat Autoimmune Diseases"

_ijms, 2020, doi:10.3390/ijms21197015_

Round 1

Reviewer 1 Report

Line 40. „maintains a state of immunological unresponsiveness to self antigens”. It is a bit superficial, since aggressive effector immune response is prevented during normal immune tolerance indeed, but tolerogenic immune response, i.e. active recognition of self-antigen and an organised anti-inflammatory, tolerogenic response is actually generated.

Line 80. Treg function as an IL2 sink is emphasized, but it is not generally accepted that this mechanism could be a major way of action of Tregs. Instead, antigen-specific direct suppressive mechanisms would deserve a few sentences, since it may have more relevance to issues discussed later, such as engineered TCR-Tregs.

Line 101: „When TCR engages with peptide-MHCII in the periphery, it is able to activate
101 the Treg by signal transduction through the TCR-CD3 complex .” This is generally valid for all types of T-cells. Could the authors mention a few mechanisms of TCR signalling that are different between Tregs and Tconvs?

Line 197: The authors point out that … to promote tolerance through selective expansion of plasmocytoid DCs”. A major role of pDCs is the production of type-I interferons, and are therefore they are regarded as major effectors in the pathomechanism of several autoimmune diseases, such as lupus. Please describe briefly how the expansion of pDSs could contribute to tolerance induction.

Line 201-208: Are there any experimental data on the modulation of TNF-receptor-2? The authors only mention the theoretical background, but if no experimental or clinical data are available, this section should be omitted.

Line 227: „However, despite clinical trials, the findings in mice have not yet been translated
228 to humans.” What is the reason? What was the conclusion of these clinical trials? A reference would be welcome.

Reviewer 2 Report

The review by Eggenhuizen et al describes the approaches to improve the Treg functions to treat autoimmune diseases. This review is written well however there is a need to describe some parts in more depth for the better understanding. Please see my specific comments.

  1. Line 9: ‘Regulatory T cells (Tregs) are a small yet critical subset of CD4+ T helper cells’. Tregs are not a subset of CD4+ T helper cells. Please correct it.
  2. For section 1, an illustration showing the phenotype and subsets of Tregs including their regulatory molecules and their effect on other immune cells, and cross talk of Tregs with other immune cells would be good for an overview of Tregs.
  3. A list of clinical trials which are using Tregs to treat autoimmune disorders with treatment regimen and outcomes could be provided if available.
  4. In DC therapy section, the provided information is really superficial. Tolerogenic DCs should be described in more depth including their phenotype, development and their role in Treg induction.
  5. The Figure 4 looks very simple and not much information can be derived from it. A detailed information like signaling molecules (surface or soluble) for each step in the figure and in the legend could be helpful for acquiring more knowledge and understanding.
  6. In the present form, the conclusion section does not look like connected with the remaining part of this review. The most important and reliable approaches to improve Tregs for the betterment of autoimmunity may be described which are mentioned in the this review. Any example of undergoing clinical trials can be taken if available.
